# The Role of Circular RNAs in Keratinocyte Carcinomas

**DOI:** 10.3390/cancers13164240

**Published:** 2021-08-23

**Authors:** Thomas Meyer, Michael Sand, Lutz Schmitz, Eggert Stockfleth

**Affiliations:** 1Department of Dermatology St. Josef Hospital, Ruhr-University Bochum, Gudrunstr. 56, 44791 Bochum, Germany; eggert.stockfleth@klinikum-bochum.de; 2Department of Plastic and Reconstructive Surgery, St. Josef-Hospital, Heidbergweg 22–24, 45257 Essen, Germany; michael.sand@ruhr-uni-bochum.de; 3Institute of Dermatopathology, MVZ Corius DermPath Bonn, GmbH, Trierer Strasse 70–72, 53115 Bonn, Germany; lutz.schmitz@dermpath-bonn.de

**Keywords:** basal cell carcinoma, cutaneous squamous cell carcinoma, circular RNA, tumourigenesis, biomarker

## Abstract

**Simple Summary:**

Keratinocyte carcinomas include BCC and cSCC and represent the most frequent cancer among fair-skinned people. Tumor development is associated with mutations and dysregulation of many genes involved in biological processes such as cell proliferation, differentiation and apoptosis. The expression of these genes is controlled in many ways, including transcriptional and post-transcriptional control by circular RNAs. In recent studies, a number of circular RNAs have been identified that are dysregulated in BCC and cSCC. Biological functions relevant to tumor development were shown for some of these circRNAs, which may represent biomarkers for disease progression and targets for novel treatment approaches.

**Abstract:**

Keratinocyte carcinomas (KC) include basal cell carcinomas (BCC) and cutaneous squamous cell carcinomas (cSCC) and represents the most common cancer in Europe and North America. Both entities are characterized by a very high mutational burden, mainly UV signature mutations. Predominately mutated genes in BCC belong to the sonic hedgehog pathway, whereas, in cSCC, *TP53, CDKN2A, NOTCH1/2* and others are most frequently mutated. In addition, the dysregulation of factors associated with epithelial to mesenchymal transition (EMT) was shown in invasive cSCC. The expression of factors associated with tumorigenesis can be controlled in several ways and include non-coding RNA molecules, such as micro RNAs (miRNA) long noncoding RNAs (lncRNA) and circular RNAs (circRNA). To update findings on circRNA in KC, we reviewed 13 papers published since 2016, identified in a PubMed search. In both BCC and cSCC, numerous circRNAs were identified that were differently expressed compared to healthy skin. Some of them were shown to target miRNAs that are also dysregulated in KC. Moreover, some studies confirmed the biological functions of individual circRNAs involved in cancer development. Thus, circRNAs may be used as biomarkers of disease and disease progression and represent potential targets of new therapeutic approaches for KC.

## 1. Introduction

Basal cell carcinomas (BCC) and cutaneous squamous cell carcinomas (cSCC) represent the vast majority of non-melanoma skin cancers (NMSC). Recently, it was suggested that these skin cancers be renamed keratinocyte cancers (KC), as they derive from epidermal keratinocytes, in contrast with other NMSC, which originate from cells other than keratinocytes (Merkel Cell Carcinoma, Lymphomas, Kaposi’s sarcoma and others) [1].

BCCs account for approximately 80% of KC and rarely metastasize or lead to death, but cause significant morbidity, due to their local infiltrative and destructive growth (locally advanced BCC) [2]. cSCC are less common than BCC, accounting for 20% of KC. However, cSCC are more aggressive than BCC. In up to 10% of patients, metastases may develop, with a high mortality rate [3,4,5].

The true numbers of KC are unclear and often underestimated, as only a few studies of incidence and mortality are available. Furthermore, many cancer registries do not register KCs or record only the first histologically confirmed KC diagnosis of every patient. The total number of NMSCs in the US population in 2012 was estimated at 5.4 million, with 3.6 million BCC and 1.8 million cSCC cases [6,7]. The incidence varies according to latitude, particularly for BCC, and is higher in southern states like Arizona and New Mexico [8]. In Europe, BCC incidence rates of around 100/100.000 inhabitants have been described [2]. The highest BCC incidence (>1.000/100.000 inhabitants) is reported in Australia [9]. However, BCC incidence is not increasing in Australia, in contrast to other continents, where BCC incidence is steadily rising [2]. Regarding cSCC, approximately 50.000 cases were estimated in Germany in 2014 [10], with men being affected more frequently. In a study comparing cSCC incidence and mortality in populations of Australia, the United States and Germany, incidence estimates in men and women of 341 and 209, 497 and 296, and 54 and 26 per 100.000 person–years were reported for the three countries, respectively [11]. The incidence increases strongly with age in all countries. Comparing cSCC incidence rates from 2002 with those from 2011–2014, the incidence was shown to decrease in Australia, whereas it is stable in the United States (2013–2015), and rising in Germany (2007 to 2015) [11]. Increasing incidence rates of BCC and SCC have also been reported in several other European countries [12,13].

As the risk of both BCC and cSCC sharply increases with older age in populations with anamnestic exposure to UV irradiation, a further increase is to be expected, based on demographic trends of an ageing population. Moreover, in the younger population, an increased BCC incidence has been noticed recently, as a result of a greater UV exposure [14].

## 2. Pathogenesis/Oncogenesis

UV radiation is the most important causative environmental factor for the development of KC. Direct absorption by nuclear DNA causes DNA damage that, if not repaired by nucleotide excision repair pathways, results in characteristic UV signature mutations (C→T or CC→TT transition) [15]. In addition, UV radiation induces a number of immune modulations that result in local and potentially systemic immunosuppression, which may impair control of dysplastic and neoplastic skin lesions [16,17]. Both BCC and cSCC are characterized by a very high mutational burden, mainly UV signature mutations, that differ in both cancer types, but clearly indicate the etiological role of UV radiation in KC development.

BCCs are thought to arise mainly from stem cells of the hair follicle [18], and are primarily driven by inactivating *PTCH1* mutations (90% of sporadic BCCs) and activating mutations of *SMO* (10% of sporadic BCC), leading to activation of the Hedgehog pathway and cell proliferation [2]. In addition, mutations of the cell-cycle regulator *TP53* are frequently detected in BCC [19]. The very few BCC cases with no mutations in the Hedgehog pathway exhibit mutations in other cancer-related genes (such as *MYCN, PPP6C, PTPN14, RB1* and *ras* oncogenes), with each detected at a low frequency in BCC, but are able to drive tumour cell transformation [20].

cSCC usually progress from precursor lesions, i.e., actinic keratosis (AK) and Bowen’s disease. Two pathways of progression of AK into invasive cSCC have been proposed. In the classical pathway, invasive SCCs occur after the involvement of the whole epidermal layer with atypical cells; in the differentiated pathway, invasion occurs directly from basal atypical keratinocytes [21]. In accordance with this, the results of a comprehensive whole-genome epigenomic analysis of cSCC development indicate that tumour cells may originate from both epidermal stem-cells and differentiated keratinocytes [22].

The most frequently mutated genes in cSCC encode the tumour suppressors p53 and p16 (*CDKN2A*). In addition, a number of other genes involved in epidermal differentiation (Notch-pathway), growth factor receptor-associated pathways (*EGFR, FGFR3*), cell-cycle control (*ras*-oncogenes), and epigenetic regulation (*KMT2C, KMT2D, TET2*) were frequently mutated, resulting in increased cell proliferation and reduced apoptosis [23].

After UV exposure, age, gender, skin type, inflammatory diseases and immunosuppression are additional important risk factors for KC development. Solid organ transplant recipients have a 60–250-fold and 6–16-fold higher risk of developing cSCC and BCC, respectively, compared to the general population [2,23,24]. For cSCC, infection with HPV is also associated with tumour development. The subgroup of β-HPV has been detected in up to 90% of cSCCs, but also in the skin of healthy controls [25]. Moreover, as viral DNA is present at low copy numbers and viral transcripts are almost absent in cSCC lesions [26], the role of β-HPV in cSCC is probably in the initiation of carcinogenesis, based on indirect mechanisms, like inhibition of DNA repair and apoptosis. Overall, the process of cSCC formation appears to involve the combined activity of several factors. UV radiation causes mutations in keratinocytes that are normally repaired by DNA repair enzymes or affected cells are eliminated by apoptosis. Both mechanisms may be impaired by β-HPV, favouring the survival of these cells and potentially leading to dysplasia and neoplasia. The resulting tumour cells will be eliminated by cellular immune functions, which, however, are impaired in immunosuppressed patients.

In contrast to BCC that rarely metastasize, cSCC may form metastases in up to 10% of cases [4]. Crossing of the basal layer is considered a key factor in invading the underlying dermis and dissemination of tumour cells. The activation of epithelial to mesenchymal transition (EMT) in basal keratinocytes represents an important mechanism supporting dermal invasion. EMT is associated with a loss of epithelial cell contacts, resulting in increased cell motility. On the molecular level, characteristic changes of the EMT program include the downregulation of epithelial markers, such as E-Cadherin and beta-Catenin, and upregulation of Vimentin, N-Cadherin, Fibronectin, MMPs and others [27,28]. Several studies have shown that EMT activation is involved in the progression of AK towards invasive cSCC [21,29,30,31,32]. The development of SCC is also associated with epigenetic processes and alterations including non-coding RNAs (miRNA, circ RNA, lnc RNA) and methylation/hydroxymethylation [33,34,35,36,37,38]. These changes affect the expression of both tumour suppressors and oncogenes and may also regulate EMT processes. The regulatory role of miRNAs in EMT and cancer has been shown in several studies [39]. For instance, miRNA hsa-miR-497, which was found to be downregulated in cSCC, counteracts EMT by regulating the expression of *SERPIN E*1 [36].

## 3. Non-Coding RNAs

A number of non-coding RNAs have been identified that are able to regulate gene expression, including micro RNAs (miRNA) long noncoding RNAs (lncRNA) and circular RNAs (circRNA). Among them, miRNAs have been studied in the most detail to date. Mature miRNAs are short endogenous RNA molecules (17–23 nucleotides in length) that are involved in the post-transcriptional regulation of gene expression, by reducing or even fully blocking the translation of specific messenger RNAs (mRNA) [40]. The biogenesis of miRNAs starts with the transcription of genomic miRNA coding sequences, which are mostly located in non-protein coding introns. The resulting primary transcripts are subsequently processed in several steps to produce mature single-stranded miRNAs that are loaded on the RNA-induced silencing complex (RISC) in the cytoplasm. Components of RISC mediate binding of the mature miRNAs to complementary sequences of the 3’untranslated region of mRNA transcripts, resulting in a decreased expression of target proteins by the degradation of transcripts or inhibition of translation, depending on the degree of complementarity of miRNA and mRNA in the seed region, which is a conserved heptametrical sequence, mostly situated at positions 2–7 from the miRNA 5′-end [41,42,43].

By modulating the expression of oncogenes and tumour suppressors, multiple miRNAs were shown to be involved in the pathogenesis of malignant tumours. Oncomir-1, a cluster of six miRNAs, also known as mir-17–92, was the first described miRNA oncogene. It regulates the expression of genes involved in cell proliferation, apoptosis and angiogenesis [44] and is associated with hematopoietic malignancies, particularly B-cell lymphomas [45]. The expression of oncomir-1 members was also significantly increased in cSCC, but not in BCC, whereas, in BCC, the expression of tumour-suppressive miRNA 145-5p was significantly decreased [35]. In addition, several other miRNAs are significantly dysregulated in cSCC and BCC compared to adjacent non-tumour control tissue. For instance, miRNA-21 is upregulated in cSCC [34] and leads to the downregulation of PDCD4 and PTEN and activation of the PI3K/Akt pathway [46,47]. On the other hand, miRNA-34a, a downstream effector of p53 involved in keratinocyte differentiation, is downregulated in cSCC [48].

lncRNAs are longer than miRNAs (>200 nucleotides) but are generally expressed at significantly lower levels [49]. Similar to miRNA, the role of lncRNAs in cellular functions was unappreciated first, and lncRNAs were regarded as non-functional or “junk” RNA molecules [50]. lncRNAs are now established as fundamental regulators of pre- and post-transcriptional gene regulation. They have been shown to be involved in cellular processes, such as the organization of nuclear sub-structures, epigenetic modifications of chromatin and regulation of gene expression. In addition, lncRNAs are able to sequester miRNAs and, upon binding to specific proteins, they may alter their subcellular localization [50,51]. In keratinocytes, specific lncRNAs are involved in the maintenance of stem cells and terminal differentiation [52]. Thus, disturbance of these keratinocyte lncRNA can result in the formation of epithelial cancers. For instance, the *HOX* transcript antisense intergenic RNA (HOTAIR) has been studied in different cancer types, including KC. It was shown to be upregulated in cSCC tissues and SCC cell lines [53,54]. In cell lines, HOTAIR promotes tumour cell proliferation, migration and EMT [54]. Proliferative and migratory effects probably result from the sequestering of miR-326, which targets PRAF2, a factor promoting cell proliferation, cell-cycle progression and cell invasion in oesophagal squamous cell carcinoma [55]. A recent RNA sequence analysis of cSCC and healthy skin using the Illumina NextSeq-500 platform identified 908 significantly different expressed lncRNAs, of which 319 were upregulated and 589 were downregulated in cSCC, including lncRNAs with oncogenic or tumour suppressor functions, described in previous studies [56]. In comparison with the microarray-based study of Sand et al. [53] 51 upregulated and 73 downregulated lncRNAs were identified in both studies, confirming the dysregulation of numerous lncRNAs associated with cSCC. The functional importance of altered lncRNA expression for tumorigenesis, however, remains to be demonstrated.

While miRNAs affect the translation of specific mRNAs, circRNAs may prevent the inhibitory effect of miRNAs on gene expression by sponging up miRNAs [57,58]. CircRNAs are covalently closed continuous loops that lack poly(A) tails and are thus less prone to degradation by cellular exonucleases [59]. Based on their origin and biogenesis, four different types of circRNAs can be distinguished: exonic circRNAs (ecircRNAs), exon-intron circRNAs (EIciRNAs), circular intronic RNAs (ciRNAs), and tRNA intronic circular RNAs (tricRNAs) [60]. The formation of most circRNA types results from back-splicing and exon skipping or lariat formation [61]. tricRNAs are synthesized from introns spliced from pre-tRNA [62]. CircRNAs can sequester miRNAs by binding to complementary sequences, called miRNA response elements (MRE), thereby acting as competing endogenous RNAs suppressing the activity of miRNAs. Thus, circRNAs, miRNAs and mRNAs form a regulatory network to control gene expression (Figure 1). Furthermore, circRNAs may directly affect the RNA transcription and protein expression of target genes by complexing RNA-binding proteins [57]. In addition, some other circRNA functions have been suggested: circRNAs may encode proteins or simply represent a negative feedback mechanism of translation, as the circRNA and linear mRNA derive from the same transcript by different splicing procedures, so that extensive back-splicing of the primary transcript to form circRNA attenuates canonical splicing to form the mature mRNA [58].

Like miRNA and lncRNA, circRNAs were also shown to be associated with the pathogenesis of different malignancies [63]. The dysregulation of numerous different circRNAs was described in various cancers, but only a rather few circRNAs were reported to be associated with the same or different types of cancer in multiple studies [58].

Similar to miRNAs, which are able to bind to a number of different mRNAs, circRNAs may target numerous different miRNAs that are complementary to the MRE. In consequence, the same miRNA or circRNA may be both tumour-suppressive and oncogenic, depending on the mRNAs or miRNAs expressed in a given cell or tissue; in other words, the role of a particular miRNA and circRNA in tumorigenesis is context-dependent. Accordingly, circular RNA circHIPK3 (hsa_circ_0000284) was shown to be upregulated in hepatocellular carcinoma (HCC), where it promotes tumorigenesis by inhibiting the tumour-suppressive activities of miR-124 [64], but it is downregulated in bladder cancer, resulting in insufficient sponging of miR-558, which promotes cancer growth through the activation of heparanase expression [65]. Thus, the same circular RNA may act as both an oncogene and tumour suppressor. Another circular RNA, hsa_circ_001569, is upregulated in colorectal cancer and HCC. It was suggested to exert pro-tumorigenic functions by sponging miR-145, a micro-RNA involved in the down-regulation of several oncogenes [66]. In BCC, the expression of miR-145-5p is also reduced compared to non-lesional skin [35], but this has not yet been linked to the increased expression of hsa_circ_001569. However, many other circular RNAs were found to be associated with BCC and cSCC. To update findings on circRNAs dysregulated in KC (cSCC and BCC) and research on their functional importance in the development of KC, we conducted a PubMed search using the terms “circular RNA” or circRNA and “cutaneous squamous cell carcinoma“ or “basal cell carcinoma”. Thirteen original papers, published between 2016 and 2021, were identified (Table 1), which are considered in the following chapters.

## 4. CircRNAs in SCC

In the first published paper on circular RNAs and cSCC by Sand et al., circRNA expression profiles were compared in six cSCC samples and six non-lesional epithelial control samples [67]. Using a circular RNA microarray containing >13.000 circRNA probes (Arraystar Human Circular RNA Microarray V2.0), 322 dysregulated circRNAs were identified (143 up- and 179 downregulated, with >2-fold different levels of expression). To predict target miRNAs, the dysregulated circRNAs were analyzed for miRNA sequences that are complementary to MREs in a bioinformatical approach. In total, 23 miRNAs were recognized, but among the top 10 up- and downregulated circRNAs, only three miRNAs were identified that interact with up-regulated circular RNAs and were reported to be differentially expressed in cSCC in previous studies: miR-9-5p and miR-124-5p both interact with hsa_circ_0035381, and miR-21-3p interacts with hsa_circ_0074817. miR-124 is involved in the regulation of extracellular signal-regulated kinase 2 (ERK2) expression and the downregulation of miR-124 in SCC cells results in abnormal cell proliferation due to the overexpression of ERK2 [79]. Thus, miR-124 may beconsidered a tumour-suppressive miRNA and sequestration by hsa_circ_0035381 could be relevant for cSCC tumourigenesis. On the other hand, the meaning of sequestering miR-9 and miR-21 by upregulated circRNAs hsa_circ_0035381 and hsa_circ_0074817 in cSCC is not clear. The expression of both miRNAs is increased in cSCC and may promote tumourigenesis [47,80]; however, it should be considered that circRNAs usually contain multiple MREs that may interact with different miRNAs, possibly including miRNA types that are not yet characterized as tumour-suppressive in cSCC.

The high expression of the three most upregulated circular RNAs in cSCC, hsa_circ_0070933, hsa_circ_0070934, and hsa_circ_0003528, was confirmed in subsequent studies [68,71,73]. These circRNAs derive from back-splicing of the primary transcript of *La Ribonucleoprotein Domain Family Member 1B* (*LARP1B*) and *SEC24 homolog A, COPII coat complex component* (*SEC24A*), respectively. Overexpression of hsa_circ_0070934 in cSCC cell lines leads to increased cell proliferation, migration and invasion, whereas knockdown inhibits cell proliferation, invasion and migration, and increases apoptosis [68,71]. Using a dual-luciferase reporter assay, An et al. showed that hsa_circ_0070934 interacts with miR-1238 and miR-1247-5p. Supplementing these miRNAs in rescue experiments supported the idea that the suppression of miR-1238 and miR-1247-5p is linked to the oncogenic role of circ_0070934, but possible mRNA targets of these miRNAs were not indicated in that study.

Zhang et al. [71] also studied the functional importance of hsa_circ_0070934 in cSCC and identified miR-1236-3p as a target molecule. In cSCC cell lines, miR-1236-3p is downregulated, resulting in impaired control of the expression of HOXB7 (homeobox B7), a transcription factor regulating several cell functions, including proliferation and EMT [81]. Accordingly, miR-1236-3p mimics inhibited the invasion and proliferation of cSCC cells and promoted apoptosis by blocking the expression of HOXB7. Transfection with circ-0070934 siRNAs also reduced the expression of HOXB7, consistent with the inhibitory effect of the miR-1236-3p mimics on HOXB7 expression levels [71]. Thus, the sequestration of miR-1236-3p by hsa_circ_0070934 may result in the abnormal proliferation, invasion and migration of cSCC cells through the increased expression of HOXB7.

Similar to hsa_circ_0070934, the knockdown of hsa_circ_0003528 also inhibits cell proliferation, invasion and migration, and induces apoptosis [73]. The authors characterized miR-1193, a tumour-suppressive miRNA that was recently shown to be down-regulated in cSCC, as a direct target of hsa_circ_0003528 [82]. Using dual-luciferase reporter assays and RNA pull-down assays, the mitogen-activated protein kinase MAP3K9, known to be associated with various cancers, was characterized as a downstream target of miR-1193. Accordingly, the same complementary miR-1193-binding sequence was found in hsa_circ_0003528 and the 3’UTR of the *MAP3K9* transcript [73]. Thus, sequestering miR-1193 by hsa_circ_0003528 promotes tumour growth through insufficient control of MAP3K9 expression, whereas inhibition of hsa_circ_0003528 suppresses cSCC tumour growth by the upregulation of miR-1193 and downregulation of MAP3K9.

Using the Arraystar Human circRNA chip, Gao et al. investigated the expression of circRNAs in cSCC tissues from three patients. In comparison to adjacent normal tissues, hsa_circRNA_001937 was increased by 14-fold. The tumour-promoting effects of hsa_circRNA_001937 were demonstrated by the circRNA knockdown in cSCC cell lines, which resulted in decreased cell proliferation and induction of apoptosis. The oncogenic functions of hsa_circ_001937 appear to include the inhibition of miRNA-597-3p, which was identified as a target miRNA that downregulates the expression of Fos-related antigen 2 (FOSL2), a member of the AP-1 transcription factor family, favouring cell proliferation [69].

Using a high-throughput sequencing approach (RNA-seq) to identify dysregulated circRNAs in cSCC, Chen et al. identified 449 circRNAs that were differentially expressed in tumour tissues and normal adjacent tissue (393 up- and 55 downregulated). Relevance for cSCC development was shown for the upregulated circRNA, circPVT1 (hsa_circ_0001821), as knockdown by RNA-silencing reduced the migration and invasion of SCL-1 cells in vitro. The authors also identified hsa_circ_0001821 as part of a circRNA–miRNA–mRNA interaction network, regulating the expression of several genes/mRNAs involved in tumourigenesis [70]. CircPVT1 (hsa_circ_0001821), is also upregulated in gastric cancer, acute myeloid leukemia and head and neck squamous cell carcinoma, and was suggested to exert pro-tumourigenic functions by sponging miR-125 and miR-497-5p [83,84]. These two miRNAs are known for their tumour-suppressive functions in cSCC, i.e., preventing EMT and migration by the downregulation of MMP13 and Serpin E1, respectively [36,85].

In another recent study from China, the expression of circRNAs was compared in three paired cSCC and adjacent non-tumour samples by RNAseq [72]. The number of 1115 dysregulated circRNAs (457 up- and 658 downregulated circRNAs) was even higher than in previous studies, indicating the great importance of differential circRNA expression in cSCC. The downregulation of two circ RNAs, hsa_circ_000932 and hsa_circ_001360, in cSCC was confirmed by RT-PCR. Experiments to functionally characterize hsa_circ_001360 in SCL-1 cells revealed that overexpression attenuated SCL-1 proliferation, invasion and migration, whereas silencing caused the opposite effects (increased proliferation, migration and invasion). Using bioinformatical tools, five miRNA targets of hsa_circ_001360 were predicted (miR-8055, miR-8063, miR-4494, miR-888-3p, and miR-8624-5p). Based on complementary sequences, these miRNAs may interact with numerous mRNAs, which may be involved in cSCC progression through different oncogenic signalling pathways, but no specific circRNA–miRNA–mRNA interactions were investigated in this study [72].

Another two circRNAs that were up-regulated in cSCC (hsa_circ_0067772 and hsa_circ-0008234) were recently functionally analyzed [74,75]. The upregulation of hsa_circ_0067772 in cSCC was shown by Sand et al. 2016 [67]. Its oncogenic function in cSCC was supported by knockdown experiments in cSCC cells that resulted in reduced cell proliferation, migration and invasion [74]. Moreover sponging miR 1238-3p circ_0067772 upregulated expression of transcription factor forkhead box protein G1 (FOXG1), which is a direct target of miR-1238-3p and acts as an oncogenic factor in ovarian cancer [86].

Cai et al. [75] used the GEO dataset GSE74758, which is based on the circRNA expression data from Sand et al. [67], to screen for differentially expressed circRNAs in cSCC. A significant upregulation of hsa_circ_0008234 was subsequently confirmed in ten paired cSCC tissues and non-lesional skin tissues by quantitative RT-PCR. Knockdown of hsa_circ_0008234 by siRNA in A431 and cSCC cells inhibited cell proliferation. It is likely that the tumour-promoting effect of hsa_circ_0008234 results from the inhibition of miR-127-5p, a tumour-suppressive miRNA, regulating the expression of a membrane-bound denylate cyclase (ADCY7), which is involved in various cellular pathways, including those associated with cancer progression [75].

In contrast with the studies mentioned above, Zhang et al. analyzed circRNAs in exosomes from cSCC patients and controls [76]. Exosomes are extracellular vesicles supporting intercellular communication by transferring key biological signal transmitters, including non-coding RNAs [87]. Of the 25 upregulated circRNAs in exosomes from cSCC circ-CYP24A1 was analyzed in functional studies. Exosomes were taken up by SCL-1 cells and the silencing of exosomal circ-CYP24A1 results in reduced proliferation, migration, and invasion, whereas apoptosis was enhanced. The authors also identified some potential downstream targets of circ-CYP24A1, but provided no further information about their possible role in tumourigenesis. This study indicates that exosomal circRNAs isolated from plasma may be used as biomarkers, which may be more accessible than lesional circRNAs.

Recently, Das Mahapatra et al. published the most comprehensive comparative transcript analysis in cSCC and healthy skin, using RNAseq of skin tissues from 18 healthy donors and 28 patients with primary cSCC [56]. Of the total 264 highly abundant circRNAs, 55 showed differential expression in cSCC and normal skin samples. Almost all of them (53/55) are downregulated in cSCC. The authors also report the upregulation of ADAR, a key negative regulator of circRNA-biogenesis, as well as the downregulation of two positive regulators of circRNA-biogenesis, MBNL and ESRP1, which might explain the general reduction in circRNAs. Nevertheless, the finding is in contrast with the studies mentioned above that report similar numbers of up- and downregulated circRNAs in cSCC. Moreover, none of the 55 dysregulated circRNAs were among the top ten dysregulated circRNAs in other studies on cSCC. However, the comparison with other studies is complicated by the fact that differences in the expression levels of >1.5-fold were considered significant, in contrast to the 2-fold changes in many other studies. In addition, some circRNAs were not included in all studies; notably, novel circRNAs cannot be detected with microarrays.

In the study of Das Mahapatra, one of the two upregulated circRNAs, circ_EPSTI1 (hsa_ circRNA_000479) was highly overexpressed in cSCC [56]. The increased expression of circ_EPSTI1 was described in many different cancers, including breast cancer, osteosarcoma, cervical cancer and ovarian cancer [88,89,90,91]. In all these studies, the silencing of circ_EPSTI1 suppressed cancer cell proliferation and invasion, but induced apoptosis or ferroptosis. The oncogenic activity of circ_EPSTI1 was shown to depend on sponging various tumour-suppressive miRNAs, which differ in all these cancers [88,89,90,91]. One of the most significantly downregulated circRNA was CDR1as (has_circ_0001964), which is also downregulated in glioma [92] but upregulated in hepatocellular carcinoma (HCC) and colorectal cancer (CRC) [93,94]. In HCC and CRC, CDR1as is considered oncogenic due to the sponging miR-7, which inhibits expression of the EGFR/RAF1/MAPK pathway, whereas, in glioma, it is potentially tumour-suppressive and the low expression levels were caused by interaction with miR-671-5p, resulting in increased tumour cell proliferation. To date, the significance of the upregulation of circ_EPSTI1 and downregulation of CDR1as for cSCC development is not known and needs to be analyzed by investigating the effects of knockdown and overexpression of these circRNAs in cSCC cell lines. To reveal functional mechanisms of the tumour progression or suppression of circ_EPSTI1 and CDR1as in cSCC, experiments using dual reporter assays or RNA pull-down assays mustbe performed to identify the potential miRNA targets in cSCC cell lines.

In the last 5 years, a number of dysregulated circRNAs were identified in cSCC in several studies. Only a few of them were found to be significantly up- or downregulated in more than one study, or were functionally characterized. The most important circRNAs, which appear to be relevant for cSCC development, are summarized in Table 2, including their target miRNAs and potential effects on tumour development.

## 5. CircRNAs in BCC

In contrast to cSCC, only two studies on circular RNA and BCC were identified in a PubMed search. The first paper, published in 2016, reports on six nodular BCC and six control samples (non-lesional epithelial skin from other subjects, but the same anatomical location) that were analyzed for circRNA expression using a microarray system (Human Circular RNA Microarray V2.0, Arraystar, MD, USA) [77]. Differential expression, defined as >2-fold change in tumour and normal specimens, was found for a total of 71 circRNAs (23 up- and 48 downregulated in BCC). Different expression levels were confirmed for three upregulated circRNAs by quantitative RT-PCR. All of the top 10 upregulated circRNAs in BCC were spliced from primary transcripts of genes previously described to be associated with different cancers. Among the four circRNAs with the highest expression in BCC, three align with *LINC00340* (alias *Cancer Susceptibility Candidate 15; CASC15*), encoding a long intergenic non-coding RNA that may function as a tumour-suppressor in neuroblastoma [95]. On the other hand, lncRNA-CASC15 has been reported as an oncogenic factor for several other cancers. For instance, in melanoma cells, knockdown of lncRNA-CASC15 inhibited proliferation, migration and invasion by inactivating Wnt/β-catenin signalling and EMT-suppression and increased cell apoptosis by downregulating Bcl-2, and survivin [96]. The importance of the *CASC15* transcripts (both the lncRNA and the circRNAs, described by Sand et al.) for BCC is not yet clear, as no functional characterization was performed.

In a bioinformatical analysis to predict miRNA-targets, Sand et al. identified MREs for various miRNAs, including several tumour-suppressive and tumour-promoting miRNAs. The top 10 up- and downregulated circRNAs contain MREs for eight and five miRNAs, respectively, previously described as dysregulated in BCC [97]. Upregulated hsa_circ_0008732 targets miR-19b-1, which is also upregulated in BCC [97] and belongs to oncomir-1, a miR cluster consisting of six miRNAs (miR-17, miR-18a, miR-19a, miR-19b-1, miR-20a, miR-92) causing increased cell proliferation and reduced apoptosis [98]. With respect to BCC development, the sequestration of a tumour-promoting miRNA by circRNAs appears paradoxical at first; however, it should be considered that circRNAs usually target a number of different miRNAs that may include both tumour-promoting and tumour-suppressive miRNAs. Furthermore, other studies suggest that many circRNAs do not just act as miRNA sponges, but may have additional functions [99].

Hsa_circ_0086419 is also upregulated in BCC and, based on the MRE sequence, may bind miR-3196a-5p. In gastric cancer, miR-3196 acts as a tumour-suppressor, as downregulation was associated with lymph node metastasis and TNM stage [100]. On the other hand, hsa_circ_0005085, which is downregulated in BCC, targets miR-93-5p and miR-106-5p, both belonging to the miR-106b-25 cluster, which may suppress apoptosis in HCC by inhibiting the expression of E2F1 transcription factor [101]. Since both miRNAs are upregulated in BCC [97], the low expression of hsa_circ-0005085 may indicate insufficient control of miR-106b and miR-93, resulting in increased tumour cell proliferation. The importance of these circRNA–miRNA interactions for BCC, however, is unclear at the present, as the functional mechanisms remain to be revealed.

The only circRNA dysregulated in BCC subjected to functional characterization is hsa_circ_0005795. Upregulation of this circRNA in BCC, initially described by Sand et al. [77], was confirmed in a recent study of tissue samples from BCC patients and cell lines (A431 and TE345.T) [78]. Increased expression of hsa_circ_0005795 was detected in 23/30 nodular BCC samples. The study also aimed to analyze the pathogenic role of hsa_circ_0005795 in BCC cell proliferation. In TE354.T cells (isolated from skin exhibiting BCC) and A431 cells (epidermoid carcinoma cell line), knockdown of hsa_circ_0005795 significantly reduced cell viability and colony formation, and promotes apoptosis. These effects may be mediated by sponging miR-1231, which was identified as a target miRNA in dual-luciferase reporter assays in both cell lines. Moreover, the silencing and overexpression of hsa_circ_0005795 increased and decreased the expression of miR-1231, respectively, in both cell lines, further supporting the oncogenic activities of hsa_circ_0005795 in BCC. However, miR-1231 was not found to be significantly downregulated in BCC tissues [97]. These findings appear somewhat conflicting, but may depend on the testing of cell lines in contrast to cancer tissue samples. As very few BCC cell lines exist (with A431 not actually a BCC cell line), attempts to study the interaction of hsa-circ_0005795 and miR-1231 in vivo may provide further evidence to confirm the relevance of miR-1231 sponging in BCC development. The cre-lox system, for instance, has successfully been used to knock down circRNAs in specific cells [102].

## 6. Future Perspectives

In both BCC and cSCC, numerous circRNAs were identified that were differently expressed compared to healthy skin, indicating an association with tumorigenesis. Many of these circRNAs result from the splicing of primary mRNAs, encoding proteins associated with malignancies, and contain MREs for miRNAs that are also dysregulated in KC, further supporting the potential involvement of these circRNAs in KC development. For BCC, the availablity of the data is still limited and further studies are required to confirm the dysregulation of particular circRNAs and identify additional circRNAs involved in tumourigenesis.

Even though the exact roles of most circRNAs in KC are still not defined, particular circRNAs may represent markers of disease and disease progression. For instance, the expression levels of circSEC24A (hsa_circ_0003528) correlate with early and advanced pathological stages of cSCC [73]. Moreover, when splitting cSCC patients into two groups with high- and low-level expression of circSEC24A, patients with high-level expression had shorter overall survival than those with low-level expression, indicating that circSEC24A might be used as a prognostic marker for cSCC [73]. In addition to disease progression, circRNAs may also predict treatment response, as shown in melanoma cell lines with high-level expression of CDR1as that were less sensitive to MAPK signalling pathway inhibitors, but significantly more sensitive to glutathione peroxidase 4 inhibitors [103]. Thus, circRNAs are potential biomarkers for treatment responses and may support the selection of treatment modalities for individual patients.

Regarding the use as a biomarker, however, it is questionable whether a single circRNA species will be sufficient to predict disease progression or treatment response, as the expression may vary individually. For instance, the upregulation of hsa_circ_0005795 in BCC, described by Sand et. al. [77] was confirmed in a subsequent study that found elevated hsa_circ_0005795 expression in most (23/30), but not all, nodular BCC samples [78]. Thus, upregulation of this circRNA appears not to be a common feature of BCC. In predicting disease progression, a panel of circRNA seems to be superior to testing individual circRNAs.

In recent studies, the silencing and overexpression of certain circRNAs revealed biological functions that are relevant for cancer development, such as the regulation of cell proliferation, migration and apoptosis. The identification of interacting miRNAs and the mRNAs targeted by these miRNAs may elucidate mechanisms associated with cSCC and BCC development. Thus, functional studies of circRNAs may reveal pathogenetic mechanisms involved in tumourigenesis. In addition, based on these functions, the stability and tissue-specific expression, circRNAs also represent promising targets of new therapeutic approaches for cSCC and BCC. CircRNAs, which are dysregulated in KC, may be targeted by gain-of-function or loss-of-function approaches, using circRNA expression plasmids, synthetic circRNAs, and RNA interference-based strategies, respectively. Indeed, circRNAs were targeted in different types of cancers, including HCC, NSCLC and gastric cancer, and the efficiency of knocking down upregulated circRNAs was shown in vitro and in vivo [104]. Recently, the potential therapeutic use of targeting specific circRNAs was also shown for cSCC in animal models. Nude mice inoculated with SCC13 cells showed significantly decreased tumour growth when circSEC24A expression in SCC13 cells was silenced by RNA interference. The knockdown of circSEC24A leads to the upregulation of miR-1193 and decreased expression of MAP3K9 in tumour tissues, indicating that silencing circSEC24A inhibited tumour progression in vivo by regulating the miR-1193/MAP3K9 axis [73]. To knockdown circRNAs in vivo, small interfering RNAs (siRNAs) or short-hairpin RNAs (shRNAs), complementary to circRNAs, are usually delivered by lipid-based vectors, such as liposomes or lipid-nanoparticles [105]. The use of gold-nanoparticles and exosomes as delivery systems may even enhance the efficiency of these RNAi-based strategies by the cell-specific delivery, improved stability and cellular up-take of molecules, as well as the reduced immune activation of siRNA and shRNA [104]. These vehicles can also be used for the delivery of circRNA expression vectors or synthetic circRNAs to increase the levels of particular circRNAs that are downregulated in cSCC or BCC. With respect to sponging miRNAs, circRNAs may be optimized by the addition of more miRNA binding sites or use of high-level expression vectors, with the aim of increasing the potency of miRNA inhibition [106]. In addition to RNA interference, the expression of circRNA can be manipulated by DNA knockout systems such as cre-lox and CRISPR/Cas systems, which have been applied to knockout specific circRNAs in vitro and in vivo [102,107,108]. To date, all treatment approaches targeting circRNAs were performed in pre-clinical studies. Although these procedures look promising, some adverse side effects, such as the unspecific silencing of other genes (off-target effects) by siRNA or shRNA, non-specific tissue targeting, the toxicity of gold-nanoparticles, or immunogenicity of interfering RNA molecules or synthetic circRNAs, need to be overcome before investigating their efficacy and safety in clinical studies.

## 7. Conclusions

As with other non-coding RNAs, the biological functions of circRNAs were first unappreciated. These RNAs are now well-known to regulate gene expression in several ways. A dysregulated expression of various circRNAs has been shown in KC (BCC and cSCC). In functional studies of some of these circRNAs, the interacting miRNAs were identified, as well as the mRNAs targeted by these miRNAs, leading to the creation of circRNA–miRNA–mRNA networks, from which pathogenetic mechanisms potentially involved in tumourigenesis may be derived. Considering the rapid progress in circRNA research and the increasing number of functionally characterized circRNAs that are dysregulated in KC, some circRNAs are promising candidates as biomarkers for disease progression and treatment response, and may also represent targets for novel therapeutic approaches. 

## Figures and Tables

**Figure 1 cancers-13-04240-f001:**
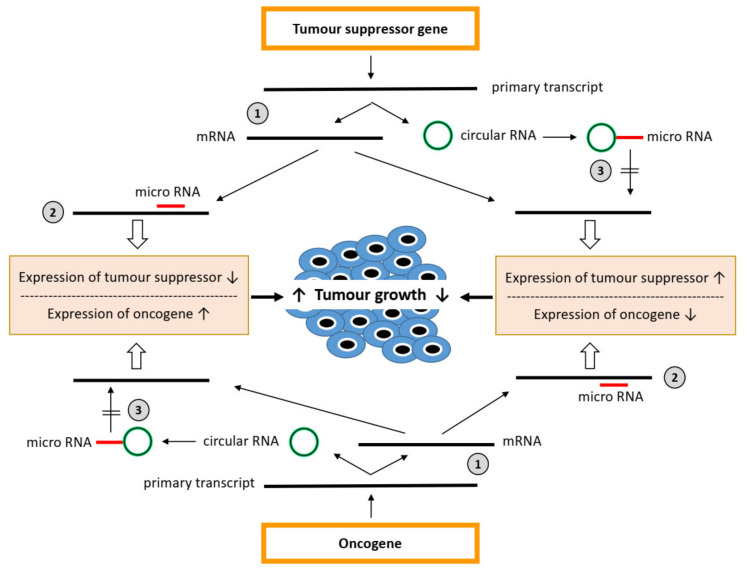
micro-RNA–circular RNA network to control mRNA expression in tumourigenesis. Expression of tumour suppressor genes and oncogenes is regulated in multiple ways, including canonical/alternative splicing of primary transcripts, resulting in mRNA and circular RNA (circRNA), respectively (1). The expression of mRNA is also regulated in multiple ways including, inhibition by micro-RNAs (miRNA) with complementary seed sequences (2). miRNAs may be inhibited by circRNAs, with specific MRE sequences sponging these miRNAs (3). They may derive from the same gene encoding the mRNA targeted by miRNAs, but also from other genes.

**Table 1 cancers-13-04240-t001:** Published articles on circular RNA and keratinocyte cancers considered in this review.

cSCC		Reference
Sand M et al. 2016	Comparison of circular RNA expression in 3 cSCC and 3 healthy control tissues by microarray analysis	[67]
An X et al. 2019	Functional characterization of hsa_circ_0070934	[68]
Gao L et al. 2020	Comparison of circular RNA expression in 3 cSCC and 3 healthy control tissues by microarray analysis and functional characterization of circRNA_001937	[69]
Chen S et al. 2020	Comparison of circRNAs in 30 paired cSCC and adjacent nontumorous tissues RNA sequencing and functional characterization of circ_PVT1	[70]
Zhang et al. 2020	Functional characterization of hsa_circ_0070934	[71]
Das Mahapatra K et al. 2020	Comparison of circRNAs, miRNAs, and lncRNAs in 9 cSCCs and 7 healthy skin samples by RNA sequencing	[56]
Chen P et al. 2021	Comparison of circRNAs in 3 paired cSCC and adjacent non-tumorous tissues RNA sequencing and functional characterization of hsa_circ_0001360	[72]
Lu X et al. 2021	Functional characterization of circSEC24A	[73]
Li X et al. 2021	Functional characterization of has_circ_0067772	[74]
Cai L et al. 2021	Functional characterization of hsa_circ_0008234	[75]
Zhang Z et al. 2021	Functional characterization of exosomal circ-CYP24A1	[76]
**BCC**		
Sand M et al. 2016	Comparison of circular RNA expression in 3 BCC and 3 healthy control tissues by microarray analysis	[77]
Li Y et al. 2021	Comparison of expression of hsa_circ_0005795 in 30 BCC and adjacent non-tumor tissues and functional characterization of hsa_circ_0005795	[78]

**Table 2 cancers-13-04240-t002:** circRNAs that are potentially relevant for cSCC and BCC development (only species confirmed in >1 study or functionally characterized in cell lines).

Circular RNA	Dysregulation in KC	Interacting miRNAs	Cellular Effects	Ref.
hsa_circ_0070933	Upregulated in cSCC	Not analyzed experimentally	Not analyzed experimentally	[67,71]
hsa_circ_0070934	Upregulated in cSCC	miR-1238, miR-1247-5p miR-1236-3p	Increased cell proliferation, migration, and invasion, inhibits apoptosis	[67,68,71]
hsa_circ_0003528 (circSEC24A)	Upregulated in cSCC	miR-1193	Increased cell proliferation, migration, and invasion, inhibits apoptosis	[73]
hsa_circ_001937	Upregulated in cSCC	miR-597-3p	Increased cell proliferation	[69]
hsa_circ_0001821 (circPVT1)	Upregulated in cSCC	Not analyzed experimentally	Increased migration and invasion	[70]
hsa_circ_001360	Downregulated in cSCC	Not analyzed experimentally	Reduced cell proliferation, migration, and invasion	[72]
hsa_circ_0067772	Upregulated in cSCC	miR-1238-3p	Increased cell proliferation, migration and invasion	[74]
hsa_circ_0008234	Upregulated in cSCC	miR-127-5p	Increased cell proliferation	[75]
Circ-CYP24A1	Upregulated in cSCC	Not analyzed	Increased cell proliferation, migration, and invasion, reduced apoptosis	[76]
hsa_circ_0005795	Upregulated in BCC	miR-1231	Increased cell proliferation, inhibits apoptosis	[78]

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
