# Peer review of "The Role of Circular RNAs in Keratinocyte Carcinomas"

_cancers, 2021, doi:10.3390/cancers13164240_

Round 1

Reviewer 1 Report

Manuscript ID: cancers-1339472

Title: The role of circular RNAs in keratinocyte carcinomas

The review of Meyer et al. summarizes findings of studies on circular RNA expression in cSCC and BCC published since 2016. In these studies, a number of circRNAs were identified with significantly different expressions in tumours and normal control tissues. Functional analysis of some of these circRNAs supports their involvement in tumour development. These studies indicate the great potential of this class of non-coding RNAs for managing cSCC and BCC, as circRNAs may be used as biomarkers for disease progression or treatment response or may even represent targets for therapeutic approaches.

I have 3 comments/suggestions which could further improve their work:

The review article is based on 10 papers identified in a PubMed search. Just two of them concern BCC. There are clearly more studies needed to confirm the data from these studies and to identify circRNAs relevant for BCC development. The authors should indicate the limited number of studies on BCC and the necessity to conduct further studies in order to identify circRNA species that are truly associated with BCC.

Regarding cSCC another 2 very recently published studies from Li et al and Cai et al. on the functional characterization of circ_0067772 and circ-0008234 in cSCC may be considered. This would improve correctness of the review paper (DOI: 10.1007/s13258-021-01074-3;DOI: org/10.1007/s00403-021-02261-8)

In functional studies of some circRNAs the interacting miRNAs were identified and also the mRNAs targeted by these miRNAs. These interactions may elucidate mechanisms associated with cSCC and BCC development. Thus, in addition, to be used as biomarkers and targets for novel therapeutic intervention, functional studies on circRNAs may also reveal pathogenetic mechanisms involved in tumorigenesis. This aspect may be inserted into the last section on future perspectives.

Author Response

The review article is based on 10 papers identified in a PubMed search. Just two of them concern BCC. There are clearly more studies needed to confirm the data from these studies and to identify circRNAs relevant for BCC development. The authors should indicate the limited number of studies on BCC and the necessity to conduct further studies in order to identify circRNA species that are truly associated with BCC.

Re: Indeed there not much data on circRNA expression in BCC and we agree with the reviewer concerning the necessity of additional studies to identfy circRNA species that are truely associated with BCC. We indicate the limited availability of data by adding the following sentence in section 6 after the first paragraph:

For BCC, availablity of data is still limited and further studies are required to confirm dysregulation of particular circRNAs and to identify additional circRNAs involved in tumourigenesis.

Regarding cSCC another 2 very recently published studies from Li et al and Cai et al. on the functional characterization of circ_0067772 and circ-0008234 in cSCC may be considered. This would improve correctness of the review paper (DOI: 10.1007/s13258-021-01074-3;DOI: org/10.1007/s00403-021-02261-8)

Re: We are greatful for this hint. In fact; by combining circRNA with cSCC we identified another paper (Zhang et al 2021), showing how rapidly the topic is evloving and we expect further studies to be published in the near future. Anayway, these additional papers are considered in the revised manuscript and also listed in table 1. Thus 13 instead of 10 papers on circRNA in cSCC are now included.

The following passages are added in section 4 after the 7. paragraph. These papers were included in the reference list and numbering of references was changed accordingly.

Another two circRNAs, up-regulated in cSCC (hsa_circ_0067772 and hsa_circ-0008234) were recently analzyed functionally (Li 2021, Cai 2021) [76,77]. Up-regulation of hsa_circ_0067772 in cSCC was shown by Sand et al. 2016 [69]. Its oncogenic function in cSCC was supported by knock-down experiments in cSCC cells that resulted in reduced cell proliferation, migration and invasion [76]. Moreover, by sponging miR 1238-3p circ_0067772 up-regulated expression of transcription factor forkhead box protein G1 (FOXG1), which is a direct target of miR-1238-3p and acts as an oncogenic factor in ovarian cancer (Chan 2009) [88].

Cai et al. [77] used the GEO dataset GSE74758, which is based on the circRNA expression data from Sand et al. [69], to screen for differentially expressed circRNAs in cSCC. Significant up-regulation of hsa_circ_0008234 was subsequently confirmed in ten paired cSCC tissues and non-lesional skin tissues by quantitative RT-PCR. Knockdown of hsa_circ_0008234 by siRNA in A431 and cSCC cells inhibited cell proliferation. Probably, the tumour promoting effect of hsa_circ_0008234 results from inhibition of miR-127-5p, a tumour-suppressive miRNA, regulating expression of a membrane-bound adenylate cyclase (ADCY7), which is involved in various cellular pathways, including those associated with cancer progression (Cai 2021) [77].

In contrast to the studies mentioned above Zhang et al. analyzed circRNAs in exosomes from cSCC patients and controls [78]. Exosomes are extracellular vesicles supporting intercellular communication by transferring key biological signal transmitters, including non-coding RNAs (Dai 2020) [89]. Of 25 circRNAs up-regulated in exosomes from cSCC circ-CYP24A1 was analyzed in functional studies. Exosomes were taken up by SCL-1 cells and silencing of exosomal circ-CYP24A1 results in reduced proliferation, migration, and invasion whereas apoptosis was enhanced. The authors also identified some potential downstream targets of circ-CYP24A1, but provide no further information about their possible role in tumourigenesis. Anyway, this study indicates that exosomal circRNAs isolated from plasma may be used as biomarkers, which may be more accessible than lesional circRNAs.

In functional studies of some circRNAs the interacting miRNAs were identified and also the mRNAs targeted by these miRNAs. These interactions may elucidate mechanisms associated with cSCC and BCC development. Thus, in addition, to be used as biomarkers and targets for novel therapeutic intervention, functional studies on circRNAs may also reveal pathogenetic mechanisms involved in tumorigenesis. This aspect may be inserted into the last section on future perspectives.

Re: We agree with the reviewers opinion and inserted the following sentences in Section 6 after the first sentence in paragraph 3:

The identification of interacting miRNAs and also the mRNAs targeted by these miRNAs may elucidate mechanisms associated with cSCC and BCC development. Thus functional studies on circRNAs may reveal pathogenetic mechanisms involved in tumourigenesis. In addition, based on these functions …

Additional papers added to the reference list:

(76) Li X., Kong Y, Li H., Xu M., Jiang M., Sun W., Xu S. CircRNA circ_0067772 aggravates the malignant progression of cutaneous squamous cell carcinoma by regulating miR-1238-3p/FOXG1 axis. Genes Genomics 2021,43,491-501. doi: 10.1007/s13258-021-01074-3.

(77) Cai L., Wang Y., Wu J., Wu G. Hsa_circ_0008234 facilitates proliferation of cutaneous squamous cell carcinoma through targeting miR-127-5p to regulate ADCY7. Arch Dermatol Res 2021, Jun 18. doi: 10.1007/s00403-021-02261-8.

(78) Zhang Z., Guo H., Yang W., Li J., Exosomal circular RNA, RNA-seq profiling and the carcinogenic role of exosomal circ-CYP24A1 in cutaneous squamous cell carcinoma. Front Med (Lausanne) 2021,8:675842. doi: 10.3389/fmed.2021.675842. eCollection 2021.

(88) Chan D.W., Liu V.W., To R.M., Chiu P.M., Lee W.Y., Yao K.M., Cheung A.N., Ngan H.Y. Overexpression of FOXG1 contributes to TGFbeta resistance through inhibition of p21WAF1/CIP1 expression in ovarian cancer. Br J Cancer 2009,101,1433–1443. doi: 10.1038/sj.bjc.6605316

(89) Dai J., Su Y., Zhong S., Cong L., Liu B., Yang J., Tao Y., He Z., Chen C., Jiang Y. Exosomes: key players in cancer and potential therapeutic strategy. Signal Transduct Target Ther 2020,5,145. doi: 10.1038/s41392-020-00261-0

Reviewer 2 Report

This is a very interesting and well written manuscript. The article is well structured and conclusions are consistent with the experimental data.

The study demonstrates that circRNAs may be used as biomarkers of disease and disease progression and represent potential targets of new therapeutic approaches for KC

The article is original as the are no similar studies published.

What specific improvements could the authors consider regarding the methodology? None 

References are appropriate.

Author Response

Thanks to the Reviewer for the positive evaluation